# Association of gout with brain reserve and vulnerability to neurodegenerative disease

Anya Topiwala [1] ✉, Kulveer Mankia[2], Steven Bell [3], Alastair Webb[4], Klaus P. Ebmeier [5], Isobel Howard [1], Chaoyue Wang [6,7], Fidel Alfaro-Almagro[6], Karla Miller [6], Stephen Burgess [8,9], Stephen Smith[6] & Thomas E. Nichols [1,6]

Studies of neurodegenerative disease risk in gout are contradictory. Relationships with neuroimaging markers of brain structure, which may offer insights, are uncertain. Here we investigated associations between gout, brain structure, and neurodegenerative disease incidence. Gout patients had smaller global and regional brain volumes and markers of higher brain iron, using both observational and genetic approaches. Participants with gout also had higher incidence of all-cause dementia, Parkinson's disease, and probable essential tremor. Risks were strongly time dependent, whereby associations with incident dementia were highest in the first 3 years after gout diagnosis. These findings suggest gout is causally related to several measures of brain structure. Lower brain reserve amongst gout patients may explain their higher vulnerability to multiple neurodegenerative diseases. Motor and cognitive impairments may affect gout patients, particularly in early years after diagnosis.

Gout is the most common inflammatory arthritis affecting ~1–4% of the population[1]. The clinical syndrome is characterised by acute flares of joint pain and swelling, resulting from deposition of monosodium urate crystals in joints and peri-articular tissues, initiating an acute inflammatory cascade. In contrast to multiple other organ systems[2], classically the brain is not thought to be affected. However, emerging studies have cited contradictory associations between hyperuricaemia and neurodegenerative disease. Observational studies have reported a lower risk of dementia, particularly Alzheimer's disease, in hyperuricaemia[3–5]. Antioxidant properties of uric acid have been proposed as a potential mechanism for this neuroprotection[6]. Mendelian randomisation studies, which can offer insights into causal relationships, have offered conflicting results in Alzheimer's disease[7,8]. Hyperuricaemia and gout have also been linked to higher stroke risk[9].

Clarifying the impact on the brain is vital given that hyperuricaemia is a treatable target.

Insights into relationships between gout and neurodegenerative disease could result from examining links with brain structure, as yet unexplored. MRI markers provide quantitative, sensitive intermediate endophenotypes for neuropsychiatric disease[10]. A few studies have investigated associations between serum urate and a handful of biomarkers for stroke and dementia. Most have found no association[11,12]. However, to date, no studies have examined gout. Urate analyses have been small ($n < 2500$), not accounted for many potential confounding variables, and explored only a few aspects of brain structure, while using solely observational approaches.

We performed the first investigation of neuroimaging markers in patients with gout. Observational and Mendelian randomisation (MR)

[1]Nuffield Department of Population Health, Big Data Institute, University of Oxford, Oxford, UK. [2]Leeds Institute of Rheumatic and Musculoskeletal Medicine, University of Leeds and NIHR Leeds Biomedical Research Centre, Leeds Teaching Hospitals NHS Trust, Chapel Allerton Hospital, Leeds, UK. [3]Department of Clinical Neurosciences, School of Clinical Medicine, University of Cambridge, Cambridge, UK. [4]Wolfson Centre for Prevention of Stroke and Dementia, Nuffield Department of Clinical Neurosciences, University of Oxford, Oxford, UK. [5]Department of Psychiatry, University of Oxford, Warneford Hospital, Oxford, UK. [6]Wellcome Centre for Integrative Neuroimaging (WIN FMRIB), Nuffield Department of Clinical Neurosciences, University of Oxford, Oxford, UK. [7]SJTU-Ruijin-UIH Institute for Medical Imaging Technology, Shanghai, China. [8]MRC Biostatistics Unit, School of Clinical Medicine, University of Cambridge, Cambridge, UK. [9]Department of Public Health and Primary Care, School of Clinical Medicine, University of Cambridge, Cambridge, UK. ✉e-mail: anya.topiwala@bdi.ox.ac.uk

approaches were combined for stronger causal inference. Furthermore, we explored relationships between gout and relevant neurodegenerative diseases. The purpose of this study was to assess whether associations between gout and brain structure would provide insights into relationships with neurodegenerative disease risk.

## Results

### Participant characteristics

In total, 11,735 participants (1165 within the imaging subset) had a diagnosis of gout. Medical professionals made the majority of diagnoses (31.1% primary care, 18.5% hospital admission). A minority (15.2%) were solely self-reported. Overall, 30.8% of gout patients reported current use of urate-lowering therapy (ULT) at assessment. Gout sufferers were older and comprised a greater proportion of males. Relationships between baseline urate and demographic variables differed by sex (Supplementary Data 1). In males, urate was positively correlated with alcohol intake and lower socioeconomic status. This was not the case in females. Mean baseline serum urate amongst asymptomatic individuals was $305.0 \pm 76.7$ μmol/L. During follow-up 3126 participants developed dementia and 16,422 individuals died. Deaths amongst gout patients were more than double those of controls (11% vs. 5%).

### Observational analyses

Gout and higher urate were associated with multiple measures of brain structure (Fig. 1). There were no marked sex differences in associations. Urate inversely associated with global brain volume (beta = −1.66E-4, $T = 3.60$, $P = 2.30$E-4), and also grey and (separately) white matter volumes, with corresponding higher cerebrospinal fluid volumes (beta=4.93E-4, $T = 4.94$, $P = 7.84$E-4). To contextualise the effect size, we related the differences to the cross-sectional effects of age (discounting non-linear effects of age). The effect of gout on the global grey matter was equivalent to a 2-year greater age, assuming all other potentially confounding factors were held constant.

There were highly significant differences in regional grey matter volumes, particularly of mid- and hindbrain structures, such as cerebellum (beta = −9.91E-04, $T = -9.26$, $P = 2.25$E-20), pons

(beta = −5.63E-04, $T = -6.23$, $P = 4.95$E-10) and midbrain (beta = −4.00 E-04, $T = -5.15$, $P = 2.67$E-07) in gout and high urate (Fig. 2). Cerebellar differences were focused on the posterior lobe, particularly inferiorly. Subcortical differences centred on the nucleus accumbens, putamen and caudate. Significant differences were also observed in white matter tract microstructure in the fornix, the major output of the hippocampus. These included lower fractional anisotropy (beta = −2.81E-4, $T = -2.83$, $P = 4.73$E-03) and higher mean diffusivity. Notably, there were no significant associations with hippocampal volume or WMH volumes. Gout and higher urate significantly associated with markers of higher iron deposition (lower T2* and higher magnetic susceptibility) of several basal ganglia structures, including bilateral putamen (beta=9.50E-04, $T = 9.25$, $P = 2.38$E-20) and caudate (7.62E-4, $T = 7.12$, $P = 1.13$E-12). Whilst associations with urate were generally robust after additional adjustment for urate-lowering therapy and consequences of hyperuricaemia, many with gout were not. Further exploration of these models clarified that blood pressure was responsible for a loss of association with cerebellar and striatal volumes, whereas markers of renal function (creatinine and cystatin C) reduced associations with basal ganglia QSM susceptibility measures. Higher urate (but not gout) was positively associated with cuneal and frontal gyrus volumes in model 1, but not in model 2 or in stratified analyses amongst solely the highest earners (Supplementary Data 2).

Gout associated with a higher incidence of dementia (average over study HR = 1.60 [1.38–1.85]). The risk was time-varying, highest in the first 3 years after gout diagnosis (HR = 7.40 [4.95–11.07]) and then decreasing (Fig. 3). We explored possible reasons for this temporal gradient. Accounting for the competing risk lowered hazard ratios but did not alter the pattern of associations (first 3 years HR = 3.53 [2.31–5.38]) (Supplementary Data 3). Excluding controls with asymptomatic hyperuricaemia (average HR = 1.59 [1.42–1.78]) or additionally controlling for possible consequences of hyperuricaemia (average HR = 1.54 [1.27–1.87]) made little impact. Receiving a gout diagnosis later in the study (independently from age) was associated with a slightly higher dementia incidence (average HR = 1.03 [1.01–1.05]). ULT at baseline assessment was not linked to dementia incidence (HR −0.25 [0.59–1.04]).

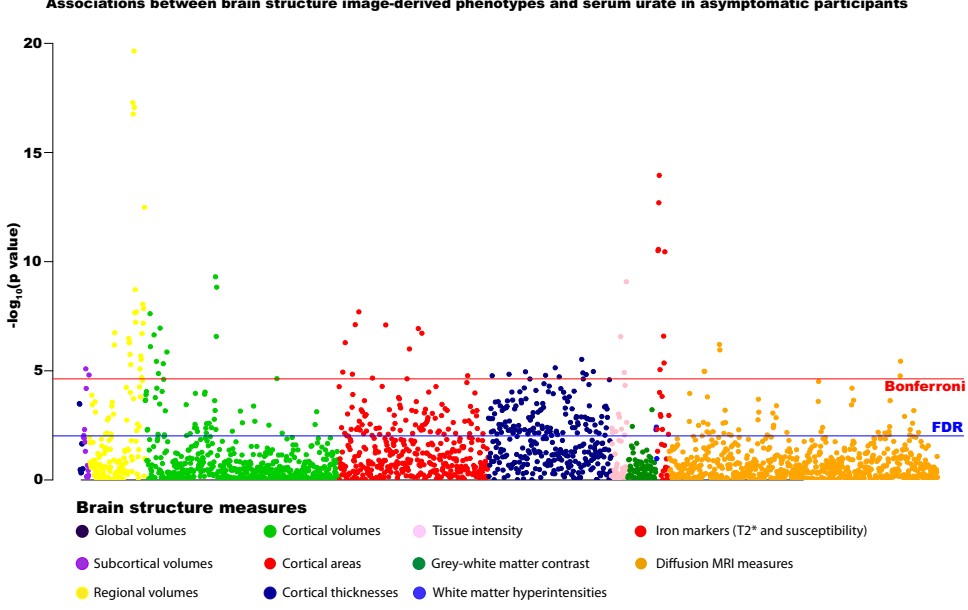

**Fig. 1 | Associations between serum urate and brain structure measures in asymptomatic participants (*n* = 33,367).** Multiple linear regression models adjusted for: all image-related confounds, age, age², sex, educational qualifications, Townsend Deprivation Index, household income, historical job code, waist-hip-ratio, alcohol intake, smoking status, diuretic use. Multiple comparison correction: blue line represents False Discovery Rate (FDR) two-sided *P* value threshold (5%) = 9.58 × 10⁻³, red line represents Bonferroni threshold on 2138 tests = 2.34 × 10⁻⁵. Source data are provided as a Source Data file.

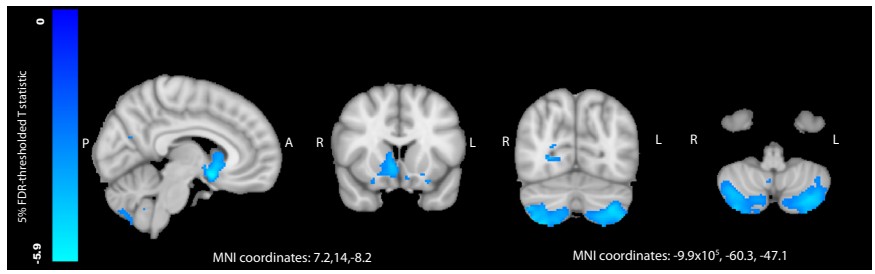

**Fig. 2 | Differences in regional grey matter volume between participants with gout (*n* = 1165) and controls (*n* = 32,202), as analysed by voxel-based morphometry.** Blue regions represent areas where participants with gout had significantly less grey matter. T statistics are thresholded at a 5% false discovery rate (0.0013 threshold on uncorrected *P* values). Models adjusted for: age, age², alcohol units weekly, imaging site, smoking status, waist-hip-ratio, total household income. FDR false discovery rate, L left, R right, A anterior, P posterior, MNI Montreal Neuroimaging Institute. Source data are provided as a Source Data file.

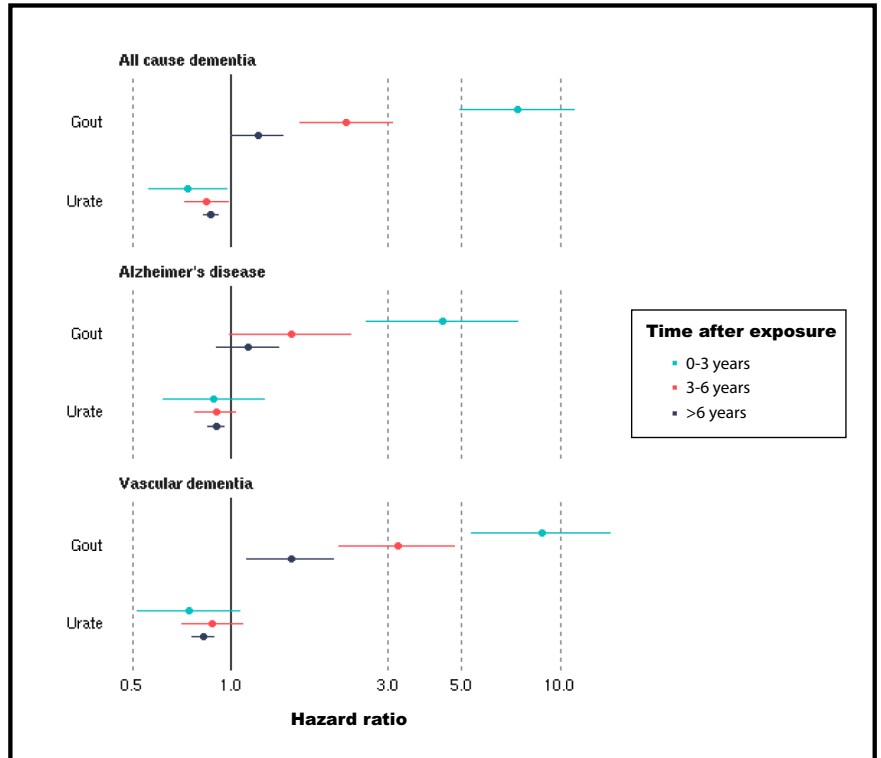

**Fig. 3 | Hazard ratios of incident dementia for gout (*N* = 303,149, 3126 dementia cases) and serum urate in asymptomatic participants (*N* = 247,328, 2454 dementia cases).** Estimates (points) and their 95% confidence intervals (bars) represent hazard ratios over different time periods after gout diagnosis or urate measurement, generated from Cox proportional hazards models, adjusted for: age, age², sex, Townsend Deprivation Index, educational qualifications, household income, historical job code, smoking, alcohol intake, waist-hip-ratio, diuretic use. Source data are provided as a Source Data file.

Risks were higher for vascular dementia (average HR = 2.41 [1.93–3.02]) compared to all-cause dementia, but not for Alzheimer's disease (average HR = 1.62 [1.30–2.02]). Again, there was a strong time dependence to the risks, particularly for vascular dementia. Amongst asymptomatic individuals, in the linear model, there was an inverse association between urate and dementia incidence (HR = 0.85, 95% CI: 0.80–0.89), with no time dependence. Using restricted cubic splines a non-linear relationship between urate and dementia incidence was observed (*P* for non-linearity <0.0001). Both low and high urate levels associated with a higher dementia incidence (Fig. 2).

Given the associations with cerebellum and striatum measures, we examined relevant phenotypes post hoc. Gout was associated with a higher incidence of both Parkinson's disease (HR = 1.43 [1.15–1.79]) and probable essential tremor (HR = 6.75 [5.59–8.00]). Amongst asymptomatic individuals, there was a linear inverse relationship between urate and incident Parkinson's disease (HR = 0.89 [0.84–0.95], *P* for non-linearity=0.3), but not with probable essential tremor (HR = 0.95 [0.89–1.02]).

## Genetic analyses

Four of the urate-associated SNPs and one of the gout-associated SNPs were unavailable in the outcome datasets. Proxy SNPs were found for three urate SNPs (Supplementary Data 4). Genetic associations mostly mirrored observational ones. Both genetically predicted gout and serum urate significantly associated with regional grey matter volumes, including cerebellar (gout IVW beta = −0.05 [−0.08 to −0.03], *P* = 1.50E-04; urate IVW beta = −0.05 [−0.09 to −0.01], *P* = 7.71E-03), midbrain (gout IVW beta = −0.07 [−0.12 to −0.03], *P* = 1.55E-03; urate IVW beta = −0.06 [−0.10 to −0.02], *P* = 7.31E-03), pons (gout IVW beta = −0.11 [−0.16 to −0.06], *P* = 3.01E-05; urate IVW beta = −0.07

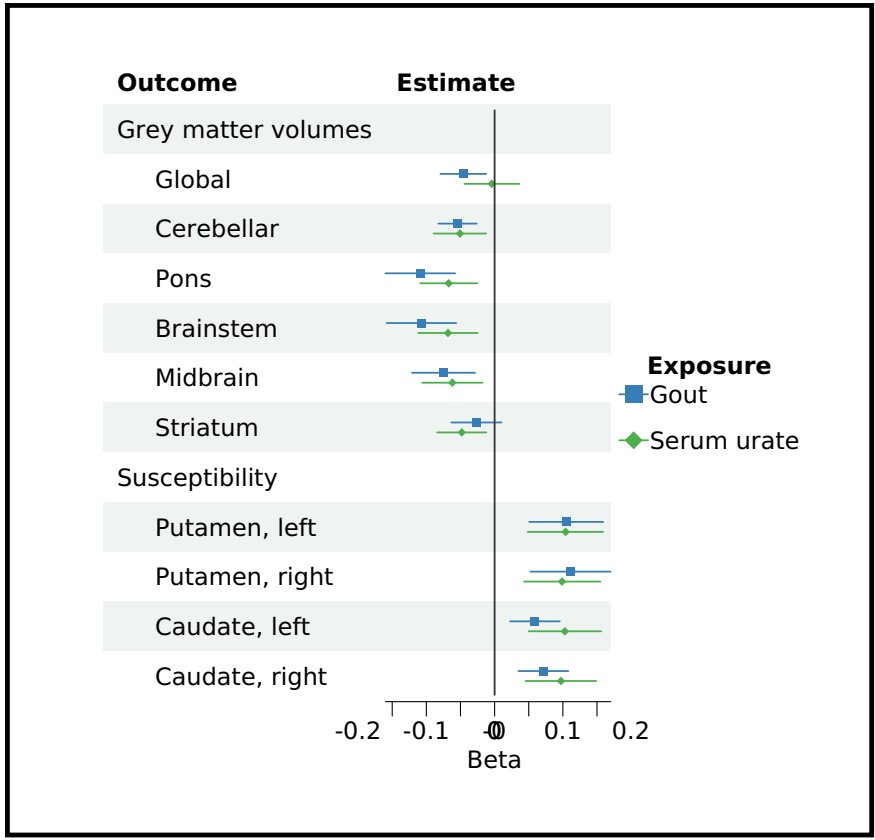

**Fig. 4 | Mendelian randomisation estimates for the association of genetically predicted gout and serum urate and brain imaging phenotypes.** Genetically predicted gout was instrumented using 12 SNPs (one-sample MR). Genetically predicted serum urate was instrumented using 182 SNPs (two-sample MR). Beta estimates (points) and their 95% confidence intervals (bars) represent inverse-variance weighted estimates for a 1 mg dl$^{-1}$ increase in serum urate, or gout versus asymptomatic hyperuricaemia. Magnetic susceptibility measures are derived from quantitative susceptibility maps, with higher values indicating higher iron. IDP image-derived phenotype, SNPs single-nucleotide polymorphism, LCI lower confidence interval, UCI upper confidence interval. Source data are provided as a Source Data file.

[−0.11 to −0.02], $P = 2.10E-03$) and brainstem (gout IVW beta = −0.11 [−0.16 to −0.06], $P = 3.78E-05$; urate IVW beta = −0.07 [−0.10 to −0.02], $P = 2.48E-03$). There were also significant associations with higher magnetic susceptibility in the putamen and caudate, markers of higher iron (Fig. 4). However, whilst genetically predicted gout was significantly associated with global grey matter volume (IVW beta = −0.05 [−0.08 to −0.01], $P = 7.75E-03$), urate did not IVW beta = −0.05 [−0.29 to −0.20], $P = 0.71$]. All associations survived FDR correction for multiple testing. Pons, brainstem and the magnetic susceptibility associations additionally survived Bonferroni correction. Robust MR methods including the weighted median and mode gave broadly consistent estimates for gout (Supplementary Data 5). However, for urate only IVW estimates were significant. Whilst there was little evidence of horizontal pleiotropy on the MR Egger intercept test, there was significant heterogeneity between IVW estimates, particularly for urate.

## Discussion

In this UK prospective cohort study, participants with a history of gout had smaller global and regional brain volumes and higher brain iron. MR analyses suggested gout was causally related to brain structure. Gout was associated with a higher incidence of several neurodegenerative diseases, particularly in the first three years after diagnosis. Lower brain reserve in gout patients may explain their vulnerability to dementia.

To our knowledge, there are no previous neuroimaging studies of gout. Investigations examining urate are small and, with two exceptions, examined solely markers of cerebrovascular disease[13]. One study reported higher white matter atrophy with hyperuricaemia[14], and another a null association with hippocampal volume[11]. We replicated these findings, but did not duplicate reported positive relationships between urate and WMH[15]. Methodological differences may be responsible. In UKB, WMH volumes were calculated automatically whereas previous studies used visual ratings. We found associations with global grey matter and cerebellar (motor and non-motor regions[16]), brainstem and striatal brain volumes. Gout patients had a higher incidence of probable essential tremor. The neurobiological correlates of essential tremor are unclear, although links have been made with cerebellar atrophy[17,18]. As this likely comprises a highly heterogeneous category we are cautious in our interpretations of these results. Associations with markers of higher iron in basal ganglia with hyperuricaemia mirror findings in aging, Alzheimer's and Parkinson's diseases[19,20], and recently alcohol consumption[21]. Our MR findings provide support for causal relationships underlying the associations. Taken together, these brain findings lend further support to recent claims of a causal impact of hyperuricaemia on Alzheimer's[7,22].

The mechanism(s) underlying how gout affects brain volume is unclear. Hyperuricaemia has been linked to arterial stiffness[23], and associated with brain microvascular damage[24], which may improve with allopurinol treatment[25]. Alternatively, toxic metabolic pathways may be responsible[26,27]. Interestingly, we found that controlling for blood pressure reduced associations between gout and brain volumes, suggesting this as a potential mediating mechanism. Whilst urate associated with measures of white matter microstructure, no relation was observed with WMHs. One explanation is that due to selection

biases, the healthier UKB imaging subset may not have yet manifested WMHs. Of note, the strongest associations of hyperuricaemia were with posterior brain regions, but the calculation of WMH volume only includes the anterior circulation. It is unclear whether, at least in healthy individuals, urate can cross the blood brain barrier[28]. High levels of iron in basal ganglia could result from inflammatory processes in gout[29,30] or poorer urinary iron excretion as suggested by our observed reduction in association after controlling for renal function. Conversely, higher ferritin (a blood measure of iron) secondary to diet, could lead to higher urate levels[29].

In this sample gout associated with higher incidence of dementia, a finding at odds with previous claims of protective effects in observational studies[13,31]. A recent small meta-analysis found no evidence for an association with gout overall, but cited a possible protective effect on Alzheimer's disease based on two studies[5]. To our knowledge, there have been just two MR studies of gout and dementia, both reporting no significant associations with Alzheimer's disease[22,32]. However, low statistical power, driven by weak instrument bias, limits their interpretation. Causal relationships with other dementia subtypes, such as vascular dementia, has been limited by the lack of available large-scale GWAS. Whilst we found gout was associated with smaller global brain volumes, associations were not observed with classical imaging markers of Alzheimer's disease[33] or vascular dementia[34] such as hippocampal volume[35] or white matter hyperintensities. Instead as gout played a causative role in multiple neurodegenerative pathologies, we propose a brain vulnerability model (Fig. 5). A similar phenomenon is well recognised with delirium[36]. Global brain volumes are commonly conceived as a proxy for brain reserve, structural characteristics of the brain that allow some people to better cope with brain pathology before cognitive changes emerge[37]. The strong temporal gradient in risk of neurodegenerative disease after gout is interesting. We propose three hypotheses. First, death is a competing risk. Second, there may be a selective detection bias. Receiving a gout diagnosis may result in more frequent medical review initially, when cognitive problems could be noticed. Third, the inflammatory response in a gout flare may induce a stepwise global decline that may precipitate acute cognitive decline.

The relationship between baseline urate and dementia incidence was non-linear. As others have previously raised, associations with a single urate measurement, which may not reflect long-term exposure[38], should be interpreted cautiously. Reduced risks of dementia with hyperuricaemia have been previously reported in observational studies[4], including in an earlier analysis of UKB[3]. Whilst earlier Mendelian randomisation studies using genetically proxied urate, reflective of lifelong exposure, were not significant[8,39], two studies published this year found support for a harmful causal effect on Alzheimer's disease[7,22]. Differing choice of SNPs as instrumental variables and outcome GWAS between studies may explain the contradictory results. Drawing together the recent MR studies and the consistent brain associations we observed in urate and gout, we hypothesise collider bias may be causing a spurious protective association for urate[40]. Both higher urate (in affluent individuals), and absence of dementia increase likelihood of UKB recruitment.

Taken together, these results lend support not only for a causal role of gout in Alzheimer's disease, but more widely in neurodegenerative disease. Furthermore, the brain associations offer a potential mechanism through which this risk is conferred. This is an area ripe for further exploration in the future with mediation MR, when large-scale GWAS of dementia subtypes become available. These findings are relevant for gout patients, clinicians and researchers. There are also wider implications for our understanding of mechanistic pathways in neurodegenerative disease that could lead to novel intervention targets. Prophylaxis for gout flares may be justified on the basis of brain health in addition to joint and cardiovascular health, although interventional trials will be needed to test this. Patients with gout should be monitored for cognitive and motor symptoms of neurodegenerative disease given their increased risk, especially in the early period after diagnosis.

## Strengths and limitations

The large size and statistical power enabled triangulation of observational and MR approaches. Consistent directions of association between different methods strengthen causal inference[41]. A comprehensive range of brain metrics were examined, whilst adjusting for multiple confounds and multiple testing. Other key strengths include the prospective cohort, examination of relevant clinical phenotypes in the same sample and time-to-event analyses accounting for competing risks.

This study has several limitations. First, UKB is a self-selective cohort[42]. Second, the measurement of serum urate was cross-sectional and may not reflect chronic exposure[43]. Third, MRI data was at a single time point so we cannot infer changes in brain structure. Fourth, case identification may be subject to ascertainment bias, likely skewed to more severe cases, which could bias associations towards the null. Fifth, there may be diagnostic confounding between Parkinson's disease and essential tremor. Sixth, gout covaries with obesity and alcohol. However, we controlled for these factors, performed MR analyses, and we have previously reported brain structure associations with alcohol in a markedly different spatial distribution in the same sample[44]. Seventh, MR relies on a set of unverifiable assumptions, which we tried to assess where possible. Whilst we found no evidence of horizontal pleiotropy (untestable Inside assumption), there was

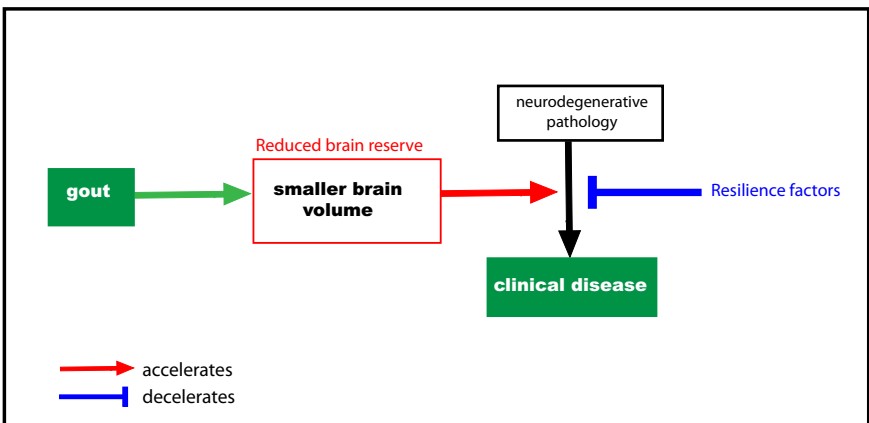

**Fig. 5 | Hypothesised model for the impact of gout on neurodegenerative disease.** Gout is causally related to smaller brain volume, a proxy for cognitive reserve. Reduced cognitive reserve indicates a reduced ability for individuals to tolerate neurodegenerative pathology before symptoms ensue.

heterogeneity of MR estimates. This may be due to failure of the assumptions of homogeneity of the genetic association with the risk factor or linearity of the effect of the risk factor on the outcome, or plausibly, differing biological causal mechanisms between variants. A minority of genetic variants used to instrument urate may have suffered from weak instrument bias. However, post hoc choice of instruments based on measured F-statistics can exacerbate bias[45]. Eighth, the strongest genetic associations were between gout and brain markers rather than urate. Sample overlap between gout and brain imaging GWAS may bias genetic associations towards observational associations. The interpretation of MR estimates with binary exposures needs care. The underlying latent continuous trait (urate), although alone is insufficient to cause gout, may influence the brain independently of gout and therefore bias MR estimates [exclusion restriction assumption][46]. As such, IV estimates from the gout MR analyses should be interpreted as an estimate for genetic compliers (likely to be a small subgroup of the population). Finally, this was not a bespoke study with optimal measures for testing cerebellar function, for example, measured gait speed or eye movements.

In this population-based sample, participants with gout had smaller global and regional grey matter volumes. Lower brain reserve may explain the vulnerability of gout patients to neurodegenerative disease. Clinicians should be vigilant for motor and cognitive problems in gout patients.

## Methods

### Study population
Participants were from UK Biobank (UKB) study[47], which recruited volunteers aged 40–69 years in 2006–2010. UK Biobank has approval from the North West Multi-centre Research Ethics Committee (MREC) as a Research Tissue Bank (RTB) approval. All participants gave informed consent. A subset underwent imaging which included brain MRI (-50,000 scanned to date). Participants underwent imaging at three centres (Newcastle, Stockport or Reading) with identical Siemens Skyra 3T scanners (software VD13) using a standard 32-channel head coil. Details of image pre-processing and quality control pipelines are described elsewhere[48]. Participants with complete data were included (Supplementary Fig. S1).

### Exposure measurements
Gout diagnoses were algorithmically defined from UKB's baseline assessment data collection, linked data from Hospital Episode Statistics, primary care and death records (https://biobank.ndph.ox.ac.uk/showcase/refer.cgi?id=460). Individuals who developed gout after dementia were excluded to minimise reverse causation. Serum urate (in µmol/L) was measured using a Beckman Coulter AU5800 at study baseline in all participants.

### Outcome measurements
**Brain markers.** We used 2138 summary image-derived phenotypes (IDPs) representing distinct measures of brain structure from the following modalities: T1-weighted and T2-weighted-FLAIR structural imaging, susceptibility-weighted MRI, diffusion MRI. These are outputs of a dedicated processing pipeline and are available from UKB[49]. IDPs included: whole brain and cerebrospinal fluid (CSF) volumes, cortical volumes, surface area, thickness, T2-FLAIR white matter hyperintensities (WMH) volumes[50], brain iron deposition metrics (T2* and magnetic susceptibility[51]), white matter microstructural measures (Supplementary Methods).

The precise spatial distribution of associations between gout and grey matter volume (T1-weighted images) across the brain was investigated using FSL-VBM[52] (Supplementary Methods), a method to compare grey matter volume in each 3D volume element (voxel) of the structural image, after adjusting between individuals for estimated total intracranial volume.

**Neurodegenerative disease.** Cases were algorithmically defined from UKB's baseline assessment data collection (including self-report), linked data from Hospital Episode Statistics, primary care and death records (https://biobank.ndph.ox.ac.uk/showcase/refer.cgi?id=460). Incident cases (diagnosed any time after baseline) were included. Prevalent cases (diagnosis before baseline) and those who developed neurodegenerative disease prior to gout were excluded, to lessen reverse causation. The primary outcome was all-cause dementia. In planned subgroup analyses, Alzheimer's and vascular dementia were examined.

### Covariates
Selected a priori confound covariates were identified on the basis of literature supporting their impact on both gout and MRI markers: all image-related confounds, age, $age^2$, sex (self-reported/NHS records), educational qualifications, Townsend Deprivation Index, household income, historical job code, waist-hip-ratio, alcohol, smoking, and diuretic use (*Model 1*). Additional adjustments for potential consequences of hyperuricaemia (potentially on the causal pathway) included: blood pressure, cholesterol, cystatin C, creatinine, diabetes mellitus, urate-lowering therapy (ULT), and chronic kidney disease (*Model 2*). Supplementary Methods describes measurement details.

### Genetic variants
Mendelian randomisation was performed for two related exposures: (1) serum urate (two-sample), and (2) gout (one-sample). Genetic instruments for urate ($n = 183$) were identified based on the sentinel variants at genome-wide significant loci ($P < 5 \times 10^{-8}$) reported in the largest publicly available genome-wide association study[53] (Supplementary Data 4). Genetic instruments for gout ($n = 13$) were identified from the largest GWAS of gout versus asymptomatic hyperuricaemia (aiming to distinguish high urate from inflammatory components of gout)[54]. Clumping was performed in the original GWAS, using a 1-Mb locus. Genetic associations with brain phenotypes were obtained from the largest GWAS of brain imaging phenotypes[55]. Proxy SNPs were selected (LD $R^2 > 0.9$) where available.

### Statistical analysis
All analyses were performed in R (version 4.1.2) unless otherwise stated.

**Observational analyses.** Multivariable linear regression models assessed the relationship between gout/urate and each of 2138 IDPs as outcomes. Brain measures and urate were quantile normalised resulting in Gaussian distributions with mean zero and standard deviation one. To account for multiple testing, Bonferroni and (separately) 5% false discovery rate (FDR) multiple comparison corrections were applied (2138 tests). Effect sizes were compared with those of age (Supplementary Methods). For VBM, the Big Linear Model toolbox (Supplementary Methods) was used to perform mass univariate ordinary least squares regression (parametric inference) voxelwise.

Cox proportional hazards models were used to estimate associations between gout/urate and neurodegenerative disease incidence. The length of follow-up was calculated as the interval between the origin and either date of event (first disease code) or censoring (date of death or last date of data collection–January 2, 2022). Origin was the baseline date, except for participants who developed gout after baseline, where the gout diagnosis date was used. The impact of time to gout diagnosis on neurodegenerative disease incidence was explored by including the interval between study baseline and gout diagnosis date as a fixed covariate. Relationships between the urate level and dementia were assessed using linear (fixed effects) and non-linear regression models. For the latter urate was categorised into quintiles and restricted cubic splines (5 knots) applied[56]. Non-linearity was formally tested (H0: $\beta2 = \beta3 = ... = \beta k - 1 = 0$) with an $F$ test.

Assumptions were checked as described in Supplementary Methods. Associations between gout and neurodegenerative disease incidence accounting for the competing risk of death were assessed using the subdistribution method[57]. In sensitivity analyses, participants with asymptomatic hyperuricaemia (defined by serum urate levels >357 μmol/L in females and >428 μmol/L in males[58]) were excluded from the control group in gout analyses.

**Genetic analyses.** We used MR to investigate whether causal relationships could underpin the observational associations we found with brain structure (Supplementary Methods for further details, including power calculation). One- (gout) and two-sample (urate) linear MR analyses using summary statistics from European participants were conducted using R packages *MendelianRandomization* (version 0.5.1) and *TwoSampleMR* (version 0.5.6). Variant harmonisation ensured that association between SNPs and exposures, and between SNPs and outcomes, reflected the same allele. Several robust MR methods were used to evaluate the consistency of the causal inference. To adjust for multiple testing, Bonferroni and false discovery rate (FDR, 5%) corrected p values were calculated across 20 tests.

**Post hoc analyses.** To investigate whether positive associations between urate and a few IDPs could be the result of residual confounding, models were stratified by income. After finding associations between gout and cerebellum, midbrain and striatum, additional relevant clinical outcomes were also examined post hoc: incidence of Parkinson's disease and incidence of other extrapyramidal movement disorders (ICD10 G25). The latter is likely dominated by essential tremor given its prevalence but also includes drug-induced tremor, myoclonus, and akathisia. We refer to this diagnosis henceforth as a probable essential tremor, although we recognise the heterogeneous nature of this entity.

### Reporting summary
Further information on research design is available in the Nature Portfolio Reporting Summary linked to this article.

## Data availability
Full pseudonymized participant data cannot be openly shared under the material transfer agreement with UK Biobank and ethics approval. Other researchers can apply for UK Biobank data to answer specific research questions. Further information about applying for data access can be obtained from the UK Biobank website (https://www. ukbiobank.ac.uk) or by emailing UK Biobank (ukbiobank@ukbiobank.ac.uk). Genetic summary statistics for serum urate, gout and brain imaging measures are freely available. Source data are provided with this paper.

## Code availability
No custom computer code or algorithm was used to generate the results reported in this paper R version 4.1.2 was used for most analyses (standard packages—survival v3.5-5, cmprsk v2.2-11, MendelianRandomization v0.7.0, TwoSampleMR v0.4.26, forester 0.5.0, ggforestplot available here: https://github.com/NightingaleHealth/ggforestplot). For VBM analyses, BLM was used (details in Supplementary Methods). Code for the VBM analyses performed is available at https://github.com/TomMaullin/BLMM.

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

## Acknowledgements

We wish to thank Maria Christodoulou and the University of Oxford Statistics Consulting service for providing advice on the survival model analyses. We also thank the Wellcome Trust—216462/Z/19/Z (A.T.), 225790/Z/22/Z (S.Bu.), 206589/Z/17/Z (A.W.), 215573/Z/19/Z (S.S.), 202788/Z/16/Z (K.Mi.), 100309/Z/12/Z (T.N.); the UK MRC—G1001354 (K.P.E.), G1001354 (K.P.E.); the European Commission—Horizon 2020 732592 (K.P.E.); the British Heart Foundation—RG/16/4/32218 (S.Be.); the NIH—R01EB026859 (T.N.); EPSRC—EP/SO2428X/1 (I.H.); and the NIHR—BRC-1215-20014 (S.Be.); NIHR203312 (S.Bu.); BRC-1215-20014 (T.N.). The funders had no role in study design, data collection and analysis, deci-sion to publish or preparation of the manuscript.

## Author contributions

A.T. conceived of the study and carried out imaging and MR analyses. C.W., F.A.A. and K.Mi. created relevant imaging data and confounds. T.N. and S.S. co-supervised the imaging analyses. S.Bu. and S.Be. co-supervised the MR analyses. K.Ma., A.W., K.P.E., I.H., S.S. and T.N. con-tributed to the interpretation of results. A.T. wrote the paper. All authors revised the paper.

## Competing interests

The authors declare no competing interests.
