## [Peer Review File · Nature Communications]

Association of Gout with Brain Reserve and Vulnerability to Neurodegenerative DiseaseREVIEWER COMMENTS

Reviewer #1 (Remarks to the Author):

This is a large scale cross sectional study of the role of urate on brain structure/volumes and impact on neurodegenerative diseases, principally alzheimers, parkinsons and a rubric called here as 'probable essential tremor.

Urate has been evaluated before in number of brain diseases and no clear picture has thus far emerged. Therefore it is timely and important that a large scale investigation such as this is undertaken. This is particularly important given that rate is modifiable by rate lowering drugs (a fact mentioned and investigated by these authors.).

In addition to direct comparisons of rate level and imaging parameters they have also used mendelian randomisation to assess potential causal relationships. This is a useful an important extra dimension to this study as it adds more to a 'simple' correlative study.

They have used well established MR approaches and they have also tried to account for reverse causation in their analyses.

I have a small number of comments/questions:

1. They speculate that the mechanisms could eb through vascular effects but if I have understood correctly they do not find any correlation with WMH (hyper intensities) - How do they explain this or any comments on this.
2. The finding related to PD is unexpected - the observational data has in the past pointed to increased urate as being protective - at least for PD risk but also possibly for progression. In fact there was a trial of a rate elevating substance a few years ago- it was stopped because of lack of impact. also there have been a small number of large scale MR studies for both risk and progression- These studies should eb cited and a discussion as to how these result conflict or influence our thinking?
3. Essential tremor diagnosis- this is a controversial issue in the field of movement disorders and almost certainly is not a uniform diagnosis. They correlate it with cerebellar volume change. It is not certain where the neuroanatomical location for ET is- again representing the heterogenous nature of this entity. Therefore I would deal and approach these data with great caution.
4. I was unclear if the imaging changes of patents with high urate or definite gout reflected the expected changes of alzheimers or FTD? or are they postulating that high rate s generally bad for brain tissue- as per their figure 5?

Reviewer #2 (Remarks to the Author):

This study uses observational (multivariable linear regression and cox proportional hazards models) and Mendelian randomization (MR) analyses for improved causal infer the causality between gout and brain structure relationships. The exposure includes gout and uric acid. GOut and serum urate is associated with regional grey matter volumes, but only gout is related to global grey matter volume, and urate did not.

My comments:

1. How can "combined observational and Mendelian randomization (MR) approaches" produce a more robust causal inference? Observational approaches in this study used multivariable linear regression and cox proportional hazards models, which cannot infer causality but correlations. The statement "the large size and statistical power enabled triangulation of observational and MR approaches for stronger causal inference" in lines 546-547 seems not very convincing.
2. This study uses two-sample MR for serum urate but one-sample MR for gout. However, the methods of MR used in this study are IVW, MR-Egger, and median MR. All methods are two-sample summarized data (using two nonoverlapping data for exposure and outcome) for MR, and thus it may serve bias for one-sample MR. Why can authors use a two-sample summarized MR in gout while both data for gout and brain imaging phenotypes come from UKB? Furthermore, the estimates of MR may be biased because gout is binary exposure (Eur J Epidemiol. 2018;33(10):947-52; Journal of the

American Statistical Association 2018; 113(522):933-947.)

3. Instrumental variables (SNPs here) play essential roles in MR. So, please provide the details of genetic variants associated with urate or gout used in this study (particularly gout), not only the summary statistics sources in STable 1. In addition, the number of genetic instruments for gout is 12, while the cited reference reported 13 independently associated genetic variants in the ABCG2, SLC2A9, SLC22A11, GCKR, MEPE, PPM1K-DT, LOC105377323, and ADH1B genes (Takei et al.; <https://ard.bmj.com/content/early/2021/06/09/annrheumdis-2021-220769>).

4. We only found the MR estimate of robust MR methods in STable 7 (lines 450-451 in the main manuscript). Even though the InSIDE assumption may be unverifiable, please discuss genetic heterogeneity and pleiotropy. For example, Is the IVW used in this study a fixed or random IVW model? How about heterogeneity test statistics? Is an SNP with associated serum urate or gout also associated with one of the brain imaging phenotypes?

5. Authors conduct univariate MR with exposure as serum urate and gout, respectively. Serum urate plays an essential role in the liability of gout. So what do authors think about conducting multivariate MR with multi-exposure (serum urate and gout) and outcome (the brain imaging phenotypes)?

Reviewer #3 (Remarks to the Author):

In this study Topiwala et al have used observational and MR methods to assess for a relationship between gout and or hyperuricaemia, reduction in volume of specific brain areas, and incidence of neurodegenerative disease. There is controversy in the literature and a large number of potential confounders (factors that are associated with both neurodegeneration and with gout/hyperuricaemia).

A significant strength of this study is that, because of the use of the UK Biobank cohort with follow up data, the observational work is prospective and less likely to be confounded by reverse causation.

Points:

- The difference interesting difference in relationship between serum urate and demographic variables between males and females; in males urate is positively correlated with alcohol intake and lower socioeconomic status. I accept that sex was included as a covariate in the later analyses but it would be interesting to see a sex stratified analysis (discussed in PMID:28665274)

- Can test statistics and effect sizes be added throughout the Results section (e.g missing from the 'Brain markers' section on p8-9) to aid interpretability.

- In the observational analyses of correlations with brain markers no measure is correlated with gout in model 2, after correction for multiple testing. This suggests that certain confounders used as covariates in model 2 may account for the signal seen in model 1. Could some effort be made to discover which covariates are responsible for the loss of power; this could add mechanistic insight which would significantly add to the manuscript impact.

- In STable 7 the Bonferroni threshold is incorrectly stated at 0.025 whereas it should be 0.0025 (20 tests)

- Too little data is presented to assess the MR results. Some of the results are unusual (e.g. MR egger tends to be underpowered vs IVW whereas this trend is reversed here). Also 1-sample MR is used here for the assessment of the relationship between gout and imaging phenotypes; this approach is vulnerable to false positive results and it is notable that this accounts for the majority of significant results here.

- Can scatter plots be provided, at least for significant results. This will aid assessment of the results.
 - Can measures such as the F-test and Cochran's Q-test be calculated to assess instrument strength and heterogeneity respectively. Also the MR egger intercept test should be performed to assess for failure of the InSIDE assumption.
 - What clumping of instrumental SNPs was performed?
 - I disagree that use of proxy SNPs is not necessary if they are available and required by missing data. Failure to do so can lead to loss of information.
- Was MR used to assess the relationship between gout / urate and incidence of neurodegenerative diseases? (as in the observational study).
- This study is missing any mechanistic insights which would significantly increase the impact. For example MVMR could be used to assess whether the effect of gout/urate on neurodegenerative diseases is mediated via the changes in brain volume.

Reviewer #1 (Remarks to the Author):

This is a large scale cross sectional study of the role of urate on brain structure/volumes and impact on neurodegenerative diseases, principally alzheimers, parkinsons and a rubric called here as 'probable essential tremor. Urate has been evaluated before in number of brain diseases and no clear picture has thus far emerged. Therefore it is timely and important that a large scale investigation such as this is undertaken. This is particularly important given that rate is modifiable by rate lowering drugs (a fact mentioned and investigated by these authors.).

In addition to direct comparisons of rate level and imaging parameters they have also used mendelian randomisation to assess potential causal relationships. This is a useful an important extra dimension to this study as it adds more to a 'simple' correlative study.

They have used well established MR approaches and they have also tried to account for reverse causation in their analyses.

I have a small number of comments/questions:

1. They speculate that the mechanisms could be through vascular effects but if I have understood correctly they do not find any correlation with WMH (hyper intensities) - How do they explain this or any comments on this.

RESPONSE: Yes this is correct, we did not find any correlation with WMH. We have three hypotheses. First, we observed significant associations with DTI measures of white matter microstructure, which likely reflect earlier and/or less severe damage. Due to selection biases, the healthier UKB imaging subset have not yet manifested WMHs. Second, the strongest associations with gout/urate were for the posterior region (as in the cerebellum), but the calculation of WMH volume only includes the anterior circulation. Third, other pathways such as toxic metabolic processes may be more relevant here than vascular effects. We have added some discussion of these hypotheses in the manuscript:

“The mechanism(s) underlying how gout affects brain volume is unclear. Hyperuricaemia has been linked to arterial stiffness [1], and associated with brain microvascular damage [2], which may improve with allopurinol treatment [3]. Alternatively, toxic metabolic pathways may be responsible [4,5]. Interestingly, we found that controlling for blood pressure reduced associations between gout and brain volumes, suggesting this as a potential mediating mechanism. Whilst urate associated with measures of white matter microstructure, no relation was observed with WMHs. One explanation is that due to selection biases, the healthier UKB imaging subset may not have yet manifested WMHs. Of note, the strongest associations of hyperuricaemia were with posterior brain regions, but the calculation of WMH volume only includes the anterior circulation.”

2. The finding related to PD is unexpected - the observational data has in the past pointed to increased urate as being protective - at least for PD risk but also possibly for progression. In fact there was a trial of a rate elevating substance a few years ago- it was stopped because of lack of impact. Also there have been a

small number of large scale MR studies for both risk and progression- These studies should be cited and a discussion as to how these result conflict or influence our thinking?

RESPONSE: Thank you for this suggestion. We now include the following discussion of this important literature:

"We found a harmful association of gout with Parkinson's disease. This is contrary to previous observational studies which suggested a protective effect of hyperuricaemia [6] which led to a trial of urate-elevating therapy as a therapeutic intervention, although this was terminated because of lack of efficacy [7]. Two MR studies of Parkinson's disease risk have found no causal effect [8,9], and two of PD progression have been contradictory [10,11]. Low statistical power was an issue in several of these studies. Taken together, we hypothesise that apparent protective associations with urate were confounded. The increased risk we observed with gout may be explained by severe or chronic hyperuricaemia or an inflammatory mechanism."

3. Essential tremor diagnosis- this is a controversial issue in the field of movement disorders and almost certainly is not a uniform diagnosis. They correlate it with cerebellar volume change. It is not certain where the neuroanatomical location for ET is- again representing the heterogenous nature of this entity. Therefore I would deal and approach these data with great caution.

RESPONSE: We agree that a cautious approach is necessary and have therefore amended the description of these *post-hoc* analyses in the methods section to emphasis this:

*"... We refer to this diagnosis henceforth as probable essential tremor **although recognise the heterogeneous nature of this entity.**"*

We have also expanded the discussion section to incorporate this concern:

*"The **neurobiological correlates of essential tremor are unclear**, although links have been made with cerebellar atrophy [12,13]. As this likely comprises a highly **heterogeneous category** we are **cautious in our interpretations** of these results."*

4. I was unclear if the imaging changes of patients with high urate or definite gout reflected the expected changes of alzheimers or FTD? or are they postulating that high rate s generally bad for brain tissue- as per their figure 5?

RESPONSE: Apologies that we did not make this clear previously. The imaging associations did not reflect classical biomarkers of Alzheimer's, vascular or frontotemporal dementia. We hypothesise instead that gout/hyperuricaemia makes the brain more vulnerable to neurodegenerative processes. We have amended the discussion to clarify this:

“Whilst we found gout was associated with smaller global brain volumes, associations were not observed with classical imaging markers of Alzheimer’s disease [14] such as hippocampal volume [15], vascular dementia markers such as white matter hyperintensities [16], nor frontal or temporal lobe volume loss as seen in frontotemporal dementia. Instead as gout appears to play a causative role in multiple neurodegenerative pathologies, we propose a brain vulnerability model (Fig 5).”

Reviewer #2 (Remarks to the Author):

This study uses observational (multivariable linear regression and cox proportional hazards models) and Mendelian randomization (MR) analyses for improved causal infer the causality between gout and brain structure relationships. The exposure includes gout and uric acid. Gout and serum urate is associated with regional grey matter volumes, but only gout is related to global grey matter volume, and urate did not.

My comments:

1. How can “combined observational and Mendelian randomization (MR) approaches” produce a more robust causal inference? Observational approaches in this study used multivariable linear regression and cox proportional hazards models, which cannot infer causality but correlations. The statement “the large size and statistical power enabled triangulation of observational and MR approaches for stronger causal inference” in lines 546-547 seems not very convincing.

RESPONSE: We agree that results from multiple linear and Cox proportional hazards regression models cannot be used to infer causality but merely provide evidence of statistical associations. However, several authors have posited that triangulation of findings from different study designs addressing the same research questions is vital to the process of making causal inferences on a topic, e.g. “*The practice of strengthening causal inferences by integrating results from several different approaches, where each approach has different (and assumed to be largely unrelated) key sources of potential bias*” [17]. We have amended the text referred to by the reviewer to clarify this point as follows:

“Consistent directions of associations between different methods strengthen causal inference [53].”

2. This study uses two-sample MR for serum urate but one-sample MR for gout. However, the methods of MR used in this study are IVW, MR-Egger, and median MR. All methods are two-sample summarized data (using two nonoverlapping data for exposure and outcome) for MR, and thus it may serve bias for one-sample MR. Why can authors use a two-sample summarized MR in gout while both data for gout and brain imaging phenotypes come from UKB? Furthermore, the estimates of MR may be biased because gout is binary exposure (Eur J

Epidemiol. 2018;33(10):947-52; Journal of the American Statistical Association 2018; 113(522):933-947.)

RESPONSE: We concur that estimates from the two-sample MR for urate are less likely to be biased than the MR for gout. However two-sample MR methods can be used safely for one-sample MR within large biobanks [18]. Whilst such an approach may suffer from biased estimates and type I error rate inflation, there is no alternative gout GWAS to our knowledge with sufficient power which excludes UKB. The reviewer is also correct in noting that, although extensively performed, there are potential issues in MR with binary exposures, and interpretation of estimates needs care. We now add discussion of both these issues to the manuscript as follows:

*“Eighth, because of GWAS availability limitations, there was sample overlap between gout and brain imaging GWAS. This may **bias genetic associations** towards observational associations and inflate type I errors.”*

*“...The interpretation of MR estimates with binary exposures needs care. The underlying latent continuous trait (urate), although alone is insufficient to cause gout, may influence the brain independently of gout and **therefore bias MR estimates** [exclusion restriction assumption] [57]. As such, IV estimates from the gout MR analyses should be interpreted as an estimate for genetic compliers (likely to be a small subgroup of the population).”*

3. Instrumental variables (SNPs here) play essential roles in MR. So, please provide the details of genetic variants associated with urate or gout used in this study (particularly gout), not only the summary statistics sources in STable 1. In addition, the number of genetic instruments for gout is 12, while the cited reference reported 13 independently associated genetic variants in the ABCG2, SLC2A9, SLC22A11, GCKR, MEPE, PPM1K-DT, LOC105377323, and ADH1B genes (Takei et al.; <https://ard.bmj.com/content/early/2021/06/09/annrheumdis-2021-220769>).

RESPONSE: We now provide in STable 1 (separate tabs for urate and gout SNPs) all SNPs used as instrumental variants (with their estimated F statistics). For urate we selected the pooled genome-wide significant variants (to maximize the amount of variance explained) but used the beta and standard errors from the European population (directionally concordant as observed in the pooled sample). The reviewer is correct in noticing that the gout GWAS we used identified 13 genome-wide significant SNPs but that we only used 12 as instruments. One SNP (rs16890979) is not available in the brain imaging MRI GWAS summary statistics [19]. We suspect this is because of slight differences between the studies in p value thresholds for SNP exclusion on the basis of Hardy Weinburg Equilibrium ($-\log_{10}(p) \leq 7$ vs. ≤ 8). Given that summary statistics for gout are only available for the SNPs achieving genome-wide significance, we are unable to use a proxy SNP with available data from both exposure and outcome datasets. We have now added SNP proxies for three (of four) urate instruments that were unavailable in the outcome dataset (as suggested by

reviewer 3), and updated the MR analyses. There was no material change to the results.

4. We only found the MR estimate of robust MR methods in STable 7 (lines 450-451 in the main manuscript). Even though the InSIDE assumption may be unverifiable, please discuss genetic heterogeneity and pleiotropy. For example, Is the IVW used in this study a fixed or random IVW model? How about heterogeneity test statistics? Is an SNP with associated serum urate or gout also associated with one of the brain imaging phenotypes?

RESPONSE: Given the large number of analyses performed we have put the full robust MR method results in STable 7, but have expanded discussion of these in the results section. We now report all heterogeneity and pleiotropy (MR-Egger intercept test) results in STable 7 as well as giving a broad overview in the results section (please see below). The IVW method used a random effects model (now stated in the methods) as generally recommended [20]. This will be equivalent to the fixed-effect analysis if there is no more heterogeneity between estimates than expected by chance, whereas in the context of excess heterogeneity (as in this case) fixed-effects analyses would give misleadingly narrow confidence intervals. None of the genetic variants used to instrument urate or gout were associated at genome-wide significance level with the brain measures.

RESULTS SECTION (Genetic associations):

“Robust MR methods including the weighted median and mode gave broadly consistent estimates for gout (STable 7) However, for urate only IVW estimates were significant. Whilst there was little evidence of horizontal pleiotropy on the MR Egger intercept test, there was significant heterogeneity between IVW estimates, particularly for urate.”

DISCUSSION (limitations):

“Whilst we found no evidence of horizontal pleiotropy (untestable inSIDE assumption), there was heterogeneity of MR estimates. This may be due to failure of the assumptions of homogeneity of the genetic association with the risk factor or linearity of the effect of the risk factor on the outcome, or plausibly, differing biological causal mechanisms between variants.”

5. Authors conduct univariate MR with exposure as serum urate and gout, respectively. Serum urate plays an essential role in the liability of gout. So what do authors think about conducting multivariate MR with multi-exposure (serum urate and gout) and outcome (the brain imaging phenotypes)?

RESPONSE: This is an interesting idea which we have puzzled over. On reflection we think a MVMR controlling for gout and urate does not make biological or clinical sense. It would mean intervening on urate levels but keeping gout risk constant, or alternatively intervening on gout whilst keeping urate levels constant. The gout GWAS we used to identify genetic variants comprised

asymptomatic hyperuricaemic individuals as controls (trying to separate out the role of non-urate inflammatory pathways to gout).

Reviewer #3 (Remarks to the Author):

In this study Topiwala et al have used observational and MR methods to assess for a relationship between gout and or hyperuricaemia, reduction in volume of specific brain areas, and incidence of neurodegenerative disease. There is controversy in the literature and a large number of potential confounders (factors that are associated with both neurodegeneration and with gout/hyperuricaemia).

A significant strength of this study is that, because of the use of the UK Biobank cohort with follow up data, the observational work is prospective and less likely to be confounded by reverse causation.

Points:

- The difference interesting difference in relationship between serum urate and demographic variables between males and females; in males urate is positively correlated with alcohol intake and lower socioeconomic status. I accept that sex was included as a covariate in the later analyses but it would be interesting to see a sex stratified analysis (discussed in PMID:28665274)

RESPONSE: Thank you for this suggestion. We have performed sex-stratified analyses of gout and urate associations with MRI markers which we now report in the manuscript (Table 3). There were no marked sex differences in associations between gout and brain measures, but we are mindful that the analyses of gout in females (n=78 cases) have much lower power to detect associations than in males (n=980). Although associations with serum urate were of a similar pattern in males and females, the latter had smaller p values for their associations with striatal, cerebellar and brainstem volumes. We think that insufficient power precludes sex stratification of the gout-dementia subtype associations. For example, there are just 79 females with gout and incident vascular dementia.

- Can test statistics and effect sizes be added throughout the Results section (e.g missing from the 'Brain markers' section on p8-9) to aid interpretability.

RESPONSE: Thank you for this suggestion. We have now added betas, T statistics and p values in several places to the results section whilst trying to keep the section readable, and refer the reader to Table 3 for the full results.

- In the observational analyses of correlations with brain markers no measure is correlated with gout in model 2, after correction for multiple testing. This suggests that certain confounders used as covariates in model 2 may account for the signal seen in model 1. Could some effort be made to discover which covariates are responsible for the loss of power; this could add mechanistic

insight which would significantly add to the manuscript impact.

RESPONSE: We would like to thank the reviewer for this suggestion which has yielded some interesting results, and we think mechanistic insights. Blood pressure appears to be responsible for the attenuation of associations between gout and cerebellar and ventral striatal volumes, whereas markers of renal function (creatinine or cystatin C) for loss of associations with QSM susceptibility (a marker of iron) in the putamen between models 1 and 2. We have added these findings to the results section of the manuscript, and comment on them further in the discussion:

“Whilst associations with urate were generally robust after additional adjustment for urate-lowering therapy and consequences of hyperuricaemia, many with gout did not. Further exploration of these models clarified that blood pressure was responsible for a loss of association with cerebellar and striatal volumes, whereas markers of renal function (creatinine and cystatin C) reduced associations with basal ganglia QSM susceptibility measures.”

“The mechanism(s) underlying how gout affects brain volume is unclear. Hyperuricaemia has been linked to arterial stiffness [1], and associated with brain microvascular damage [2], which may improve with allopurinol treatment [3]. Alternatively, toxic metabolic pathways may be responsible [4,5]. Interestingly, we found that controlling for blood pressure reduced associations between gout and brain volumes, suggesting this as a mediating mechanism..... High levels of iron in basal ganglia could result from inflammatory processes in gout [21,22], poorer urinary iron excretion as suggested by our observed reduction in association after controlling for renal function. Conversely, higher ferritin (a blood measure of iron) secondary to diet, could lead to higher urate levels [21]. ”

- In STable 7 the Bonferroni threshold is incorrectly stated at 0.025 whereas it should be 0.0025 (20 tests)

RESPONSE: Thank you for spotting this typological error which has been corrected. We have also amended the FDR threshold after updating the analyses to include 3 proxy SNPs for urate (as discussed below).

- Too little data is presented to assess the MR results. Some of the results are unusual (e.g. MR Egger tends to be underpowered vs IVW whereas this trend is reversed here). Also 1-sample MR is used here for the assessment of the relationship between gout and imaging phenotypes; this approach is vulnerable to false positive results and it is notable that this accounts for the majority of significant results here.

RESPONSE: We now give more data in the supplementary materials for the MR results, including heterogeneity and pleiotropy (MR-Egger intercept test) results (STable 7), and scatterplots (SFigure 3). We also now discuss these results in the manuscript results and discussion sections. With the exception of gout on striatal volume, IVW estimates are stronger than MR Egger estimates, as expected by the lower power of MR Egger. We agree it is an important point that the most

significant MR results are with genetically predicted gout, and that 1-sample MR can be biased towards the observational associations. However others have found that two-sample MR methods can be used safely for one-sample MR within large biobanks [18]. We have highlighted this as a limitation in the discussion as follows:

“Eighth, the strongest genetic associations were between gout and brain markers rather than urate. Sample overlap between gout and brain imaging GWAS may bias genetic associations towards observational associations.”

- Can scatter plots be provided, at least for significant results. This will aid assessment of the results.

RESPONSE: Scatterplots are now provided for all MR analyses in SFigure 3.

- Can measures such as the F-test and Cochran's Q-test be calculated to assess instrument strength and heterogeneity respectively. Also the MR Egger intercept test should be performed to assess for failure of the InSIDE assumption.

RESPONSE: We have now estimated F statistics for all genetic variants used as instruments (STable 1). SNPs used to instrument urate were identified from the largest pooled GWAS (to maximize the amount of variance explained) but used the beta and standard errors from the European population (directionally concordant) [23]. Although some instruments have low F statistics, de-selecting variants based on measured instrument strength in the dataset under analysis induces more bias than you remove. As instrument strength is estimated, and so if you remove at a given threshold, then you disproportionately include lucky variants (that is, variants where the association/strength was by chance overestimated) and exclude unlucky variants (that is, variants where the association/strength was by chance underestimated). As stated in (<https://pubmed.ncbi.nlm.nih.gov/21414999/>): *“Post hoc choice of instruments, genetic models or data based on measured F-statistics can exacerbate bias. In particular, the commonly cited rule of thumb that $F > 10$ avoids bias in IV analysis is misleading.”* We now report results from heterogeneity and MR Egger intercept tests (STable 7), and discuss these in the results and discussion section, for example:

“Whilst there was little evidence of horizontal pleiotropy on the MR Egger intercept test, there was significant heterogeneity between IVW estimates, particularly for urate.”

“... we found no evidence of horizontal pleiotropy (untestable InSIDE assumption), there was heterogeneity of MR estimates. This may be due to failure of the assumptions of homogeneity of the genetic association with the risk factor or linearity of the effect of the risk factor on the outcome, or plausibly, differing biological causal mechanisms between variants. A minority of genetic variants used to instrument urate may have suffered from weak instrument bias. However post hoc choice of instruments based on measured F-statistics can exacerbate bias [24].”

- What clumping of instrumental SNPs was performed?

RESPONSE: Instrumental variant SNPs used selected on the basis of those achieving genome-wide significance in the original GWAS. Clumping was performed by authors of the original GWAS we did not repeat this. We have amended the methods to include details of this:

"Clumping was performed in the original GWAS, using a 1-Mb locus."

- I disagree that use of proxy SNPs is not necessary if they are available and required by missing data. Failure to do so can lead to loss of information.

RESPONSE: We agree that, where possible, proxy SNPs can be informative and apologise for the clumsy wording previously regarding this point. Four urate SNPs were missing from the outcome dataset, of which we have found available proxies for three (detailed in STable 1). We have updated the MR analyses accordingly (there were no material differences in the results). For the gout MR, the only instrumental variable (SNP) that was not available in the outcome data was rs16890979 in the gout analyses. Unfortunately full summary statistics from the gout GWAS are not publically available, so we were unable to find a proxy SNP. We now detail how proxy SNPs were selected and where they were used:

METHODS:

"Proxy SNPs were selected ($LD R^2 > 0.9$) where available."

RESULTS:

"Four of the urate-associated SNPs and one of the gout-associated SNPs were unavailable in the outcome datasets. Proxy SNPs were found for three urate SNPs (STable 1)."

- Was MR used to assess the relationship between gout / urate and incidence of neurodegenerative diseases? (as in the observational study).

RESPONSE: We agree it would be very interesting to directly compare observational and MR estimates for neurodegenerative disease. Unfortunately the current lack of availability of large GWAS for the most relevant phenotypes, such as vascular dementia, or essential tremor, precludes this currently. There have been recent MR studies of urate and Alzheimer's [25,26] and Parkinson's disease [3-6], so we do not think there is a strong rationale to repeat these. We now include a statement to this effect in the limitations.

"Causal relationships with other dementia subtypes, such as vascular dementia, has been limited by the lack of available large-scale GWAS."

- This study is missing any mechanistic insights which would significantly increase the impact. For example MVMR could be used to assess whether the effect of gout/urate on neurodegenerative diseases is mediated via the changes in brain volume.

RESPONSE: Mediation analyses would be very interesting. Along this line of thinking, we did perform causal mediation analyses in UKB imaging subsample, and found that whilst the direct effect of gout on incident dementia was not significant, there was suggestion of mediation through total grey matter volume ($\beta = 2.46 \times 10^{-5}$, CI: 7.53×10^{-5} to 0.00). However, as the number of dementia cases within this sample was extremely small ($n=65$), we decided the power was too low to be confident about this result so have not included it in the manuscript. In terms of multivariable/mediation MR analyses, as discussed above, there is currently unfortunately no large-scale vascular dementia GWAS, which would be the most relevant clinical phenotype. Furthermore, in the observational analyses the dementia risk appears strongly time-dependent which current MR methods would not be able to detect. We have added the following to the discussion:

“This is an area ripe for further exploration in the future with mediation MR, when large-scale GWAS of dementia subtypes become available.”

1. Albu A, Para I, Porojan M. Uric acid and arterial stiffness. *Therapeutics and clinical risk management*. 2020;16:39.
2. Mitchell GF, Van Buchem MA, Sigurdsson S, Gotlib JD, Jonsdottir MK, Kjartansson Ó, et al. Arterial stiffness, pressure and flow pulsatility and brain structure and function: the Age, Gene/Environment Susceptibility–Reykjavik study. *Brain*. 2011;134(11):3398-407.
3. Ng KP, Stringer SJ, Jesky MD, Yadav P, Athwal R, Dutton M, et al. Allopurinol is an independent determinant of improved arterial stiffness in chronic kidney disease: a cross-sectional study. *PLoS One*. 2014;9(3):e91961.
4. Pierce DR, Williams DK, Light KE. Purkinje cell vulnerability to developmental ethanol exposure in the rat cerebellum. *Alcoholism: Clinical and Experimental Research*. 1999;23(10):1650-9.
5. Manto M. Toxic agents causing cerebellar ataxias. *Handbook of clinical neurology*. 2012;103:201-13.
6. Weisskopf M, O'reilly E, Chen H, Schwarzschild M, Ascherio A. Plasma urate and risk of Parkinson's disease. *American journal of epidemiology*. 2007;166(5):561-7.
7. Bluett B, Togasaki DM, Mihaila D, Evatt M, Rezak M, Jain S, et al. Effect of urate-elevating inosine on early Parkinson disease progression: the SURE-PD3 randomized clinical trial. *JAMA*. 2021;326(10):926-39.
8. Kobylecki CJ, Nordestgaard BG, Afzal S. Plasma urate and risk of Parkinson's disease: a Mendelian randomization study. *Annals of Neurology*. 2018;84(2):178-90.
9. Kia DA, Noyce AJ, White J, Speed D, Nicolas A, Collaborators I, et al. Mendelian randomization study shows no causal relationship between circulating urate levels and Parkinson's disease. *Annals of neurology*. 2018;84(2):191-9.
10. Simon KC, Eberly S, Gao X, Oakes D, Tanner CM, Shoulson I, et al. Mendelian randomization of serum urate and parkinson disease progression. *Annals of neurology*. 2014;76(6):862-8.

11. Coney R, Storm CS, Kia DA, Almramhi M, Wood NW. Mendelian Randomisation Finds No Causal Association between Urate and Parkinson's Disease Progression. *Movement Disorders*. 2021;36(9):2182-7.
12. Bagepally BS, Bhatt MD, Chandran V, Saini J, Bharath RD, Vasudev M, et al. Decrease in cerebral and cerebellar gray matter in essential tremor: A voxel-based morphometric analysis under 3T MRI. *Journal of Neuroimaging*. 2012;22(3):275-8.
13. Pietracupa S, Bologna M, Tommasin S, Berardelli A, Pantano P. The contribution of neuroimaging to the understanding of essential tremor pathophysiology: a systematic review. *The Cerebellum*. 2021:1-23.
14. Mckhann GM, Knopman DS, Chertkow H, Hyman BT, Jack Jr CR, Kawas CH, et al. The diagnosis of dementia due to Alzheimer's disease: Recommendations from the National Institute on Aging-Alzheimer's Association workgroups on diagnostic guidelines for Alzheimer's disease. *Alzheimer's & dementia*. 2011;7(3):263-9.
15. Duguid J, De La Paz R, Degroot J. Magnetic resonance imaging of the midbrain in Parkinson's disease. *Annals of neurology*. 1986;20(6):744-7.
16. Wardlaw JM, Valdés Hernández MC, Muñoz-Maniega S. What are white matter hyperintensities made of? Relevance to vascular cognitive impairment. *Journal of the American Heart Association*. 2015;4(6):e001140.
17. Munafò MR, Davey Smith G. Robust research needs many lines of evidence. *Nature*. 2018;553(7689):399-401.
18. Minelli C, Del Greco M F, Van Der Plaats DA, Bowden J, Sheehan NA, Thompson J. The use of two-sample methods for Mendelian randomization analyses on single large datasets. *International journal of epidemiology*. 2021;50(5):1651-9.
19. Smith SM, Douaud G, Chen W, Hanayik T, Alfaro-Almagro F, Sharp K, et al. An expanded set of genome-wide association studies of brain imaging phenotypes in UK Biobank. *Nature neuroscience*. 2021;24(5):737-45.
20. Burgess S, Smith GD, Davies NM, Dudbridge F, Gill D, Glymour MM, et al. Guidelines for performing Mendelian randomization investigations. *Wellcome Open Research*. 2019;4.
21. Fatima T, Mckinney C, Major TJ, Stamp LK, Dalbeth N, Iverson C, et al. The relationship between ferritin and urate levels and risk of gout. *Arthritis research & therapy*. 2018;20(1):1-9.
22. Yuan S, Larsson S. Causal associations of iron status with gout and rheumatoid arthritis, but not with inflammatory bowel disease. *Clinical Nutrition*. 2020;39(10):3119-24.
23. Tin A, Marten J, Halperin Kuhns VL, Li Y, Wuttke M, Kirsten H, et al. Target genes, variants, tissues and transcriptional pathways influencing human serum urate levels. *Nature genetics*. 2019;51(10):1459-74.
24. Burgess S, Thompson SG, Collaboration CCG. Avoiding bias from weak instruments in Mendelian randomization studies. *International journal of epidemiology*. 2011;40(3):755-64.
25. Şanlı BA, Whittaker KJ, Motsi GK, Shen E, Julian TH, Cooper-Knock J. Unbiased metabolome screen links serum urate to risk of Alzheimer's disease. *Neurobiology of Aging*. 2022;120:167-76.

26. Ou Y-N, Zhao B, Fu Y, Sheng Z-H, Gao P-Y, Tan L, et al. The Association of Serum Uric Acid Level, Gout, and Alzheimer's Disease: A Bidirectional Mendelian Randomization Study. *Journal of Alzheimer's Disease*. 2022(Preprint):1-11.

REVIEWER COMMENTS

Reviewer #1 (Remarks to the Author):

The authors have clarified many of the points and I thank them for the efforts they have made. I also agree that combining differing approaches, providing they all point in the same direction of effect, does mitigate some of the concerns. But there does remain some worries about the potential confounders/biases of 1 stage MR data for gout estimates. I do agree that they cannot do much about this given the size of GWA needed (UK biobank) to generate the instruments. I am also somewhat reassured that the 2 sample urate estimates (which one can reasonably assume mirrors and closely aligns with gout also point in the same direction.

Overall I think they have done a good job on the revisions.

Reviewer #2 (Remarks to the Author):

The authors' response is satisfactory. I have no further comments.

Reviewer #3 (Remarks to the Author):

The authors have added several new analyses in response to Reviewer comments. More detail is provided re the MR work but it would be useful if the authors could add effect sizes and p-values to the main Results section, as they did for the observational work. The scatter plots should be added to the main figures.

With opportunity to review the robust MR tests and heterogeneity test results I think the serum urate MR should be removed. The Q-test reveals significant instrument heterogeneity suggesting that the IVW estimate is unreliable. As the authors state in the Results section, the serum urate MR relies almost entirely on the IVW result; for serum urate (unlike the gout MR) there are no significant robust analyses (after multiple testing correction) to suggest that the IVW result is a true positive.

Reviewer #1

The authors have clarified many of the points and I thank them for the efforts they have made. I also agree that combining differing approaches, providing they all point in the same direction of effect, does mitigate some of the concerns. But there does remain some worries about the potential confounders/biases of 1 stage MR data from gout estimates. I do agree that they cannot do much about this given the size of GWAS needed (UK biobank) to generate the instruments. I am also somewhat reassured that the 2 sample urate estimates (which one can reasonably assume mirrors and closely aligns with gout also point in the same direction.

Overall I think they have done a good job on the revisions.

Thank you.

Reviewer #2

The authors' response is satisfactory. I have no further comments.

Thank you.

Reviewer #3

The authors have added several new analyses in response to Reviewer comments. More detail is provided re the MR work but it would be useful if the authors could add effect sizes and p-values to the main Results section, as they did for the observational work.

RESPONSE: We have now added several effect sizes and p values to the MR results section. Robust results we have kept in STable 7 as listing all these in the main text makes the paragraph difficult to read.

The scatter plots should be added to the main figures.

RESPONSE: There are 16 scatter plots in total which we feel would be better suited to supplementary rather than main figures, but we leave this to the editors' discretion.

With opportunity to review the robust MR tests and heterogeneity test results I think the serum urate MR should be removed. The Q-test reveals significant instrument heterogeneity suggesting that the IVW estimate is unreliable. As the authors state in the Results section, the serum urate MR relies almost entirely on the IVW result; for serum urate (unlike the gout MR) there are no significant robust analyses (after multiple testing correction) to suggest that the IVW result is a true positive.

RESPONSE: In the interests of full scientific transparency, we feel it is important not to select results that support our case and suppress those which do not. Furthermore, as reviewer 1 points out, there is a distinct advantage of including the two-sample urate MR together with the one-sample gout MR. We state clearly in the results section that robust methods for urate were not significant, and all these results are presented in STable 7. Significant heterogeneity of IVW causal estimates for urate (as assessed by Cochran's Q statistic) could be the result of several factors in addition to possible violation in the IV assumptions. These include

multiple mechanisms of intervention on a complex risk factor, each of which has an associated causal effect (plausible in the case of urate).